# Comprehensive infectious disease screening in a cohort of unaccompanied refugee minors in Germany from 2016 to 2017: A cross-sectional study

Ales Janda[1¤], Kristin Eder[1], Roland Fressle[2], Anne Geweniger[1], Natalie Diffloth[1], Maximilian Heeg[1,3], Nadine Binder[4,5], Ana-Gabriela Sitaru[6], Jan Rohr[1,3], Philipp Henneke[1,3], Markus Hufnagel[1], Roland Elling[1,3,7] *

1 Department of Pediatrics and Adolescent Medicine, Medical Center—University of Freiburg, Freiburg, Germany, 2 Practice for Childhood and Adolescent Medicine, Freiburg, Germany, 3 Institute for Immunodeficiency, Center for Chronic Immunodeficiency, Medical Center—University of Freiburg, Faculty of Medicine, University of Freiburg, Freiburg, Germany, 4 Institute for Prevention and Cancer Epidemiology, Faculty of Medicine and Medical Center, University of Freiburg, Freiburg, Germany, 5 Institute of Digitalization in Medicine, Faculty of Medicine and Medical Center, University of Freiburg, Germany, 6 Center of Laboratory Diagnostics, MVZ Clotten, Freiburg, Germany, 7 Berta Ottenstein Programme, University Medical Center, Medical Faculty, University of Freiburg, Freiburg, Germany

¤ Current address: Department of Pediatrics and Adolescent Medicine, University Medical Center Ulm, Germany

* roland.elling@uniklinik-freiburg.de

## Abstract

### Background

Information regarding the prevalence of infectious diseases (IDs) in child and adolescent refugees in Europe is scarce. Here, we evaluate a standardized ID screening protocol in a cohort of unaccompanied refugee minors (URMs) in a municipal region of southwest Germany.

### Methods and findings

From January 2016 to December 2017, we employed a structured questionnaire to screen a cohort of 890 URMs. Collecting sociodemographic information and medical history, we also performed a standardized diagnostics panel, including complete blood count, urine status, microbial stool testing, tuberculosis (TB) screening, and serologies for hepatitis B virus (HBV) and human immunodeficiency virus (HIV). The mean age was 16.2 years; 94.0% were male, and 93.6% originated from an African country. The most common health complaints were dental problems (66.0%). The single most frequent ID was scabies (14.2%). Of the 776 URMs originating from high-prevalence countries, 7.7% and 0.4% tested positive for HBV and HIV, respectively. Nineteen pathogens were detected in a total of 119 stool samples (16.0% positivity), with intestinal schistosomiasis being the most frequent pathogen (6.7%). Blood eosinophilia proved to be a nonspecific criterion for the detection of parasitic infections. Active pulmonary TB was identified in 1.7% of URMs screened. Of note, clinical warning symptoms (fever, cough >2 weeks, and weight loss) were insensitive parameters

**Data Availability Statement:** All relevant data are within the manuscript and its Supporting Information files.

**Funding:** The authors received no specific funding for this work.

**Competing interests:** The authors have declared that no competing interests exist.

**Abbreviations:** BMI, body mass index; CXR, chest X-ray; EAP, European Academy of Paediatrics; EASL, European Association for the Study of the Liver; EBUS-TBNA, endobronchial ultrasound-guided transbronchial needle aspiration; HAV, hepatitis A virus; HBsAg, hepatitis B antigen; HBeAg, hepatitis B e-antigen; HBV, hepatitis B virus; HIV, human immunodeficiency virus; ID, infectious disease; IgG, immunoglobulin G; IGRA, interferon gamma release assay; LTBI, latent TB infection; MCV, median corpuscular volume; MHC, median hemoglobin concentration; MIF, methiolate–iodine–formalin; STROBE, Strengthening the Reporting of Observational Studies in Epidemiology; TB, tuberculosis; URM, unaccompanied refugee minor..

for the identification of patients with active TB. Study limitations include the possibility of an incomplete eosinophilia workup (as no parasite serologies or malaria diagnostics were performed), as well as the inherent selection bias in our cohort because refugee populations differ across Europe.

## Conclusions

Our study found that standardized ID screening in a URM cohort was practicable and helped collection of relevant patient data in a thorough and time-effective manner. However, screening practices need to be ameliorated, especially in relation to testing for parasitic infections. Most importantly, we found that only a minority of infections were able to be detected clinically. This underscores the importance of active surveillance of IDs among refugees.

## Author summary

### Why was this study done?

- Unaccompanied refugee minors (URMs)—refugees under 18 years of age who migrate without being accompanied by a parent or a custodian—belong to the most vulnerable subgroup of refugees.

- Little is known about the prevalence and clinical presentation of infectious diseases (IDs) among pediatric refugees or URMs in particular.

- A better understanding of the frequency and clinical presentation of IDs among minor refugees is of high priority in order to improve their healthcare and develop more effective ID screening strategies for this population.

### What did the researchers do and find?

- We evaluated a systematic ID screening algorithm for refugee minors among a cohort of 890 URMs in a municipal area located in southwest Germany during 2016–2017.

- We found scabies to be the most prevalent ID (present in 14.2% of URMs), whereas active tuberculosis and human immunodeficiency virus (HIV) infection were of relatively low prevalence (1.7% and 0.4%, respectively).

- Parasite screening through serial stool investigations of patients with eosinophilia had a low diagnostic yield yet was associated with significant costs and logistic challenges.

- In general, most of the diagnosed chronic infections among URMs were not detected clinically.

### What do these findings mean?

- Among refugees, ID screening needs to be performed independently of clinical complaints because most infections cause nonspecific symptoms or are asymptomatic.

- Moving forward, ID screening among refugees urgently needs to be standardized across Europe, including the implementation of digital health records that are easily accessible to healthcare providers across all transit countries.

## Introduction

In 2015, the European Union began experiencing a significant influx in refugees, especially from Africa and Asia. In the years since, ongoing migration has posed important economic, political, and healthcare challenges to the continent. Because the endemicities of many infectious diseases (IDs) vary globally, it seems reasonable to screen refugees for IDs that may be more prevalent in those coming from low- and middle-income countries. This includes infections with *Mycobacterium tuberculosis*, hepatitis B virus (HBV), human immunodeficiency virus (HIV), and parasitic diseases. Despite the potential for transmission, it is generally accepted that infected individuals do not represent a significant risk to populations in host countries [1,2]. Nevertheless, chronic infections negatively impact refugees' general well-being while also increasing their morbidity and mortality. Furthermore, chronic parasitic infections may lead to additional problems such as anemia, nutrient deficiency, and stunting [3]. Often, the chronic and oligosymptomatic course of many IDs found in migrants leads to delays in diagnosis—delays compounded by language and cultural barriers, as well as by limited access to care. For these reasons, these diseases require active screening.

Most European countries have a low incidence of tuberculosis (TB; i.e., <20 cases/100,000 inhabitants/year). The majority of refugees, however, originate from low-income countries with high TB incidence (i.e., >100 cases/100,000 inhabitants/year). This is particularly an issue in the sub-Saharan region. In Gabon, for instance, TB incidence is approximately 60 times higher than it is in Germany [4]. The prevalence of chronic HBV infection in Africa overall (6.1%) is approximately four times higher than it is in Europe (1.5%) [5]. In relation to Germany (0.3%), the contrast with Africa is even more striking, with prevalence in Africa being 20-fold higher than it is in Germany [5,6]. Africa also has a high prevalence of HIV infection, (4.1% on average; >10% in some sub-Saharan countries), whereas European prevalence is approximately 0.4% (and, in Germany, 0.2%) [7].

During the peak period of 2015–2017, Germany had the highest immigration rate in Europe [8]. During this time, 1,444,225 asylum applications were filed with German authorities. Every fourth applicant was <18 years old, and among this group, 67,275 were entering the country as unaccompanied refugee minors (URMs), i.e., without their parents or other family members [8]. These child and adolescent refugees, exposed to hardships during travel via land or sea, represent a particularly vulnerable population. They are especially at risk for psychological trauma, malnutrition, and ID.

No systematic ID surveillance system for refugees exists in either Europe or Germany. Information on the ID burden among refugee minors, particularly URMs, is scarce. A few cross-sectional studies recently have been conducted in Germany [9–14]. However, none of them has adequately covered the abovementioned spectrum of ID in a URM cohort. Some of these reports have focused on urine and stool parasites [9–12], whereas others have focused on chronic hepatitis B prevalence [10–12] or on TB [13]. Three studies addressed a fuller range of relevant issues, but they included only a limited number of participants (102 and 154 URMs, respectively) [11,14], or else they focused on a select URM population (e.g., only Syrian refugees) [12].

In contrast to other world regions [15–17], until very recently, there were no EU-wide guidelines for ID screening of refugee minors. A review of international guidelines with experts' recommendations from the European Academy of Paediatrics (EAP) only was released in August 2019 [18]. In 2015, a consensus paper on screening recommendations for refugee minors in Germany was published [19]. Due to lack of epidemiological data on ID in this group, screening recommendations were mainly based upon advice gathered from ID experts.

The goal of the study was to evaluate the performance and practicability of these screening recommendations [19]. We carried out a single-center, 2-year screening study of a cohort of URMs in southwest Germany.

## Methods

### Study design, reporting, and prespecified analysis plan

This 2-year (2016–2017) cross-sectional study followed the Strengthening the Reporting of Observational Studies in Epidemiology (STROBE) reporting guideline (S1 STROBE Check-list). The study's prespecified analysis plan consisted of an algorithm for screening based upon the country of origin and clinical symptoms of each refugee minor (Fig 1).

### Screening process

In order to standardize the screening protocol based upon history taking, physical examination, and laboratory investigations among URMs, we developed an electronic questionnaire to be used by the screening physician. The resulting data were directly importable into electronic database programs (S1 Text). This approach provided a structured, consistent format for patient screening and history taking while also facilitating data extraction and detailed data analysis. Screening was conducted at a single private pediatric practice in Freiburg, Germany. Here, in addition to pediatric care for the general population, a select team of pediatricians offered specific consultation hours for URMs. If a specific infection such as active TB was suspected during the initial visit, then the diagnostic workup was broadened to include non-prespecified analyses (e.g., chest CT scan). In accordance with German recommendations for the screening of refugee minors, all screening results subsequently were reviewed by pediatric ID specialists [19]. Referred through the regional reception center for URMs, patients were accompanied by a social worker and an interpreter. The study period lasted 24 months: from January 1, 2016, through December 31, 2017. Patients with pathological screening results were referred to the Pediatric Infectious Disease Department at the Freiburg University Medical Center for further evaluation. The structured patient history questionnaire included sociodemographic questions such as duration of transit, transit countries and route, and languages spoken, along with a set of medical history questions focused on underlying chronic conditions, drug use, present complaints, clinical signs of TB, and gastrointestinal infections. Mental health concerns were not systematically explored during the screening. However, if an underlying mental health condition was suspected during the general history taking or via indirect questions (e.g., sleep problems, mood instability, signs of anxiety, suicide intentions, or substance use), then patients were referred to the Department of Child and Adolescent Psychiatry at the Freiburg University Medical Center for additional psychological or psychiatric evaluation. The physical exam included a complete physical examination, as well as a basic vision test and audiometry. All patients underwent a full blood count in order to identify anemia, systemic inflammation, and eosinophilia (cutoff >500 eosinophils/μl) indicative of parasite infections [19].

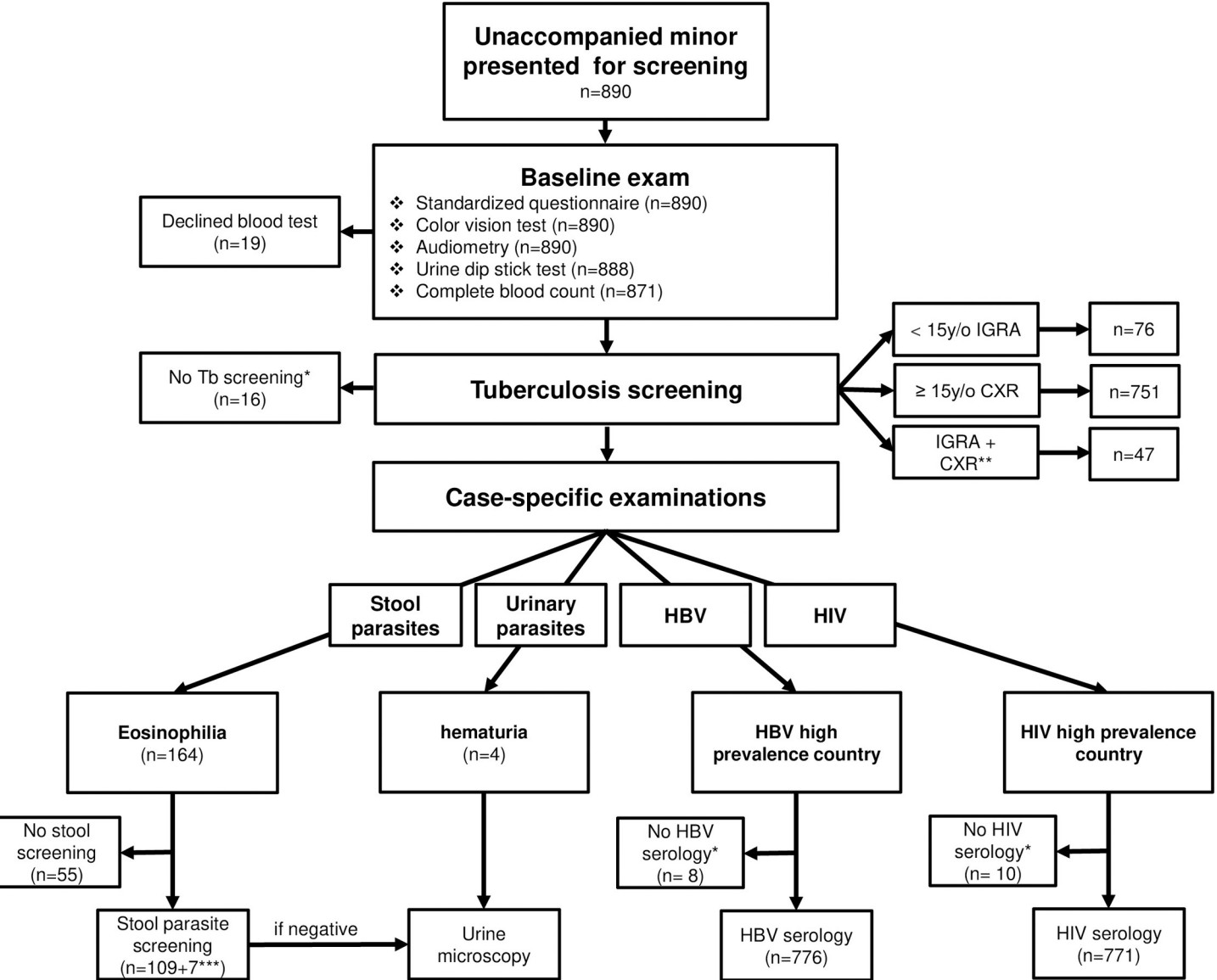

**Fig 1. Flowchart of applied screening algorithm in a cohort of unaccompanied refugee minors (*n* = 890).** *Participants refused screening or previously already screened; **comprehensive initial diagnostics in symptomatic patients (fever, cough >2 weeks, loss of weight); ***patients with gastrointestinal symptoms. *Based on Pfeil and colleagues [19]*. CXR, chest X-ray; HBV, hepatitis B virus; HIV, human immunodeficiency virus; IGRA, interferon gamma release assay; TB, tuberculosis.

All patients diagnosed with somatic diseases other than IDs were referred for specialized care and appropriately treated, if indicated. This also was the case for URMs who were transferred to another region of Germany. In instances of HIV or HBV positivity, special attention was given to follow-up across regions. Nonmedical aspects of care for refugees such as education and translation were provided through the help of social workers closely looking after the refugees at the URM housing facilities.

## Vaccination strategy

None of the URMs in our Freiburg cohort were able to bring immunization records with them. They therefore were considered to be vaccination-naive. Reimmunization according to

the German recommendations for missed immunizations [19] was started on the day of screening. All URMs received a first dose of MMRV and DTPaIPV vaccines. During the winter months, URMs also received a dose of influenza vaccine at the time of the screening visit. Follow-up vaccinations (two doses of MMRV and three doses of DTPaIPV in total), as well as vaccinations against HBV (three doses), pneumococcus (one dose), and HPV (two to three doses, depending upon age), were deferred to the pediatricians/general practitioners who took over the URMs' medical care in the communities in which the URMs later settled.

## Laboratory examinations

In patients with eosinophilia or gastrointestinal symptoms, stool tests for parasites and helminths with a target sample of three independent stool samples per patient were performed. Fecal samples were diluted in saline and stained with methiolate–iodine–formalin (MIF; Parasite Concentration System, BioRepair). Ova, cysts, trophozoites, and adult worms were identified by their characteristic microscopic features. All patients received a urine dipstick test, primarily for the purpose of identifying microhematuria as an indicator for urogenital schistosomiasis. Serologies for HIV and HBV were performed in patients from high-prevalence countries (HIV: prevalence in country of origin of $\geq$1%; HBV: prevalence in country of origin of $\geq$8%) [5,7,19]. For HIV infection, HIV-1 and HIV-2 antibody and p24 antigen chemiluminescence microparticle immunoassays (CMIA, ACHITECT System, Abbott) were used as screening tests, and HIV-1 and HIV-2 antibody western blot as a confirmatory test. HBV screening included hepatitis B surface antigen and anti–hepatitis B surface antibodies (CMIA, ARCHITECT System, Abbott). Serology for hepatitis C infection was performed only in patients with HBV infection or else in cases of clinical suspicion. General TB screening was performed on an age-dependent basis. In children and adolescents $\leq$15 years of age, an immunodiagnostic screening with an interferon gamma release assay (IGRA) was conducted. In accordance with national German requirements [19,20], in all patients >15 years of age, a chest X-ray was performed, regardless of symptoms [19,20]. The planning, conduct, and reporting are in accordance with the Declaration of Helsinki, as revised in 2013.

## Ethics

A legal guardian was assigned to each URM by the municipality that provided consent for the required diagnostic and therapeutic measurements. In a majority of cases, a registered interpreter informed the URMs about the planned healthcare checkup with the study. Because of the high rate of URM illiteracy, formal written consent was not obtained. In the rare instances when an interpreter was not immediately available, URMs were provided information by a person who spoke a language that the URM understood. Healthcare coverage and appropriate treatment of identified diseases were made possible through government-provided health insurance. This allowed URMs to obtain treatment comparable to that received by nonimmigrant children in Germany. Data analysis was able to be performed in an anonymized fashion because the data points from the electronic questionnaire were exported into a data spreadsheet without patient identifiers. The study protocol was submitted to the ethics committee of the University of Freiburg as a noninterventional clinical practice study, and approval was granted after detailed discussion (study identifier 340/18).

## Statistical analyses and graphics

Statistical analyses were performed using R (v 3.4.4) and the dplyr (v 0.7.8) package. Graphics were generated using the open source R packages ggplot2 (v 3.1.0) and maps (v 3.3.0) for the map in Fig 2C, as well as Adobe Illustrator CS6, Microsoft Excel, and GraphPad Prism

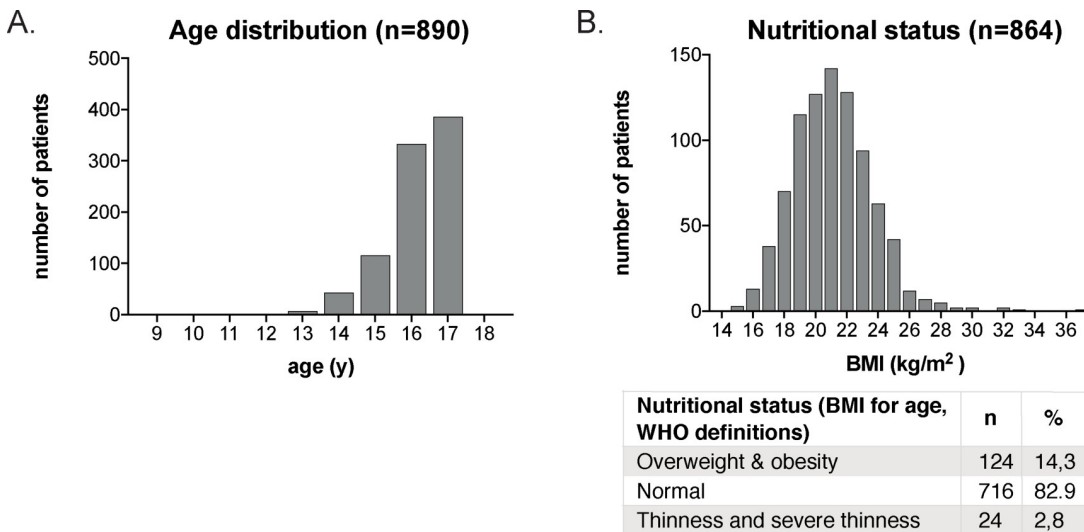

A. Age distribution (n=890)

B. Nutritional status (n=864)

| Nutritional status (BMI for age, WHO definitions) | n | % |
|---|---|---|
| Overweight & obesity | 124 | 14,3 |
| Normal | 716 | 82.9 |
| Thinness and severe thinness | 24 | 2,8 |

C. Country of origin (n=887) and route to Europe (n=721)

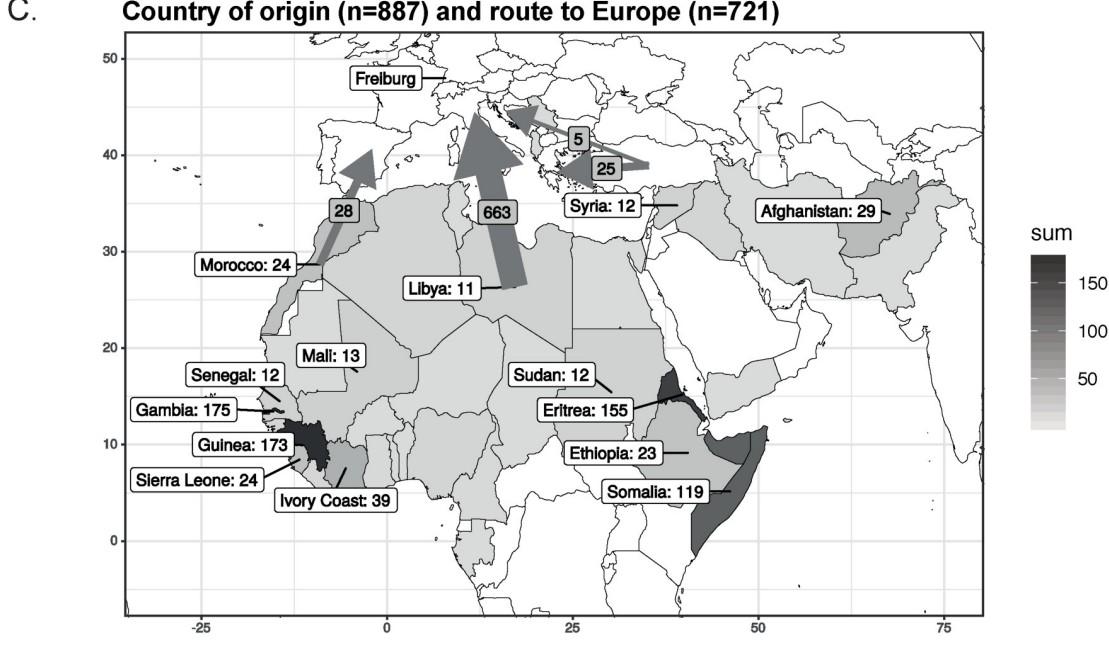

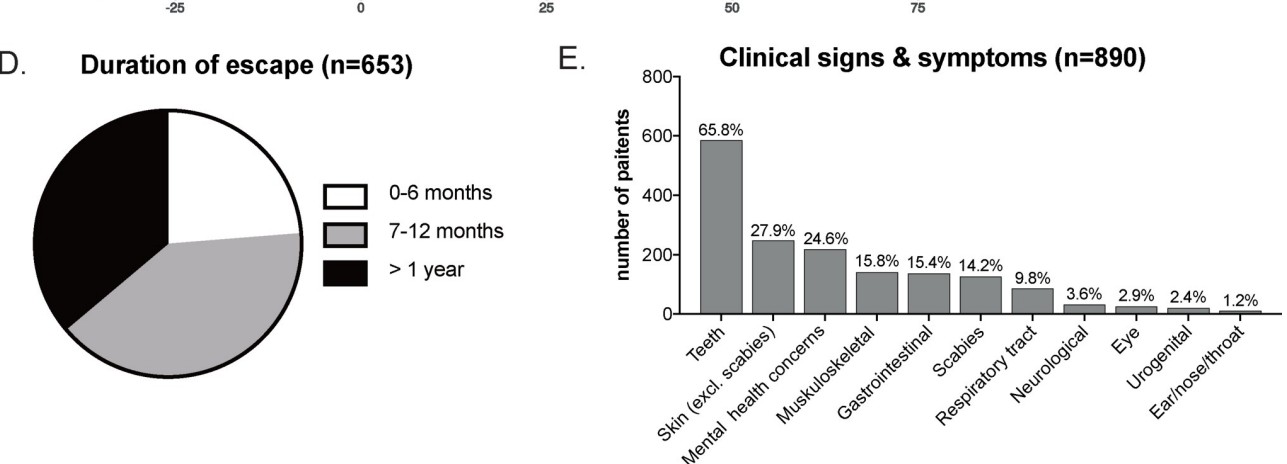

D. Duration of escape (n=653)

- 0-6 months
- 7-12 months
- > 1 year

E. Clinical signs & symptoms (n=890)

**Fig 2. Sociodemographic characteristics of the screened cohort of unaccompanied refugee minors (*n* = 890).** (A) Age distribution; (B) nutritional status; (C) route and (D) duration of migration, as well as country of origin; (E) clinical signs and symptoms. The gray scale indicates the frequency of refugees from a specific country (see legend on the right). The size of the arrow indicates the relative frequency of chosen route to Europe, with the central Mediterranean route being the most important transit route during the study period. BMI, body mass index.

(Version 8). Comparison across patient subgroups was carried out using two-sample *t* test for continuous variables and chi-square tests for categorical variables. *P* values < 0.05 were considered significant.

## Definition of variables

In relation to migration routes, URMs followed one of four main migration paths: the (1) western Mediterranean, (2) central Mediterranean, (3) eastern Mediterranean (Aegean Sea), or via land through the (4) Balkan route. Reference values for body mass index (BMI) were in accordance with those published by WHO [21]. Age-related local reference values for hemoglobin, median corpuscular volume (MCV), median hemoglobin concentration (MHC), and transaminase enzyme activity (ASAT, ALAT) were applied. The pathological threshold for eosinophils was set at 500 cells/μl.

## Results

### Sociodemographic data

In total, 890 URMs were screened following the algorithm described in the Methods section (Fig 1). Most were adolescents between 16 and 17 years old (80.8%, median age 16.2 years, IQR 15–17 years); 5.8% (*n* = 52) were <15 years old (Fig 2A). The stated date of birth must, however, be interpreted with caution because most refugees did not possess any personal identification documents, and statements they provided could not be independently verified. BMI as an indicator of nutritional status was known in 864 URMs (97.1%) and, in a majority of cases, was normal (82.9%; Fig 2B). Overweight status and obesity were more prevalent than underweight status (14.3% versus 2.8% [21]). The vast majority of URMs were male; only 6.0% (*n* = 54) of the cohort were female. The URMs originated from 35 different countries. As shown in Fig 2C (arrows) and S1 Table, 93.6% of the patients came from Africa (*n* = 830), whereas 6.0% (*n* = 53) came from Asia, and just a small number (*n* = 4, 0.5%) from southern Europe. The central Mediterranean route was the one most frequently used to reach Europe (*n* = 663, 92.0%), followed by the western Mediterranean route (*n* = 28, 3.9%) and the eastern Mediterranean route across the Aegean Sea (*n* = 25, 3.5%). Only five URMs entered Europe by land (Balkan route). Information on the duration of travel from the country of origin to Germany was available for 653 participants (Fig 2C). Over two-thirds of all URMs arrived after a minimum of 6 months of travel. The median time from the country of origin to Germany was 12 months (range 1–96 months; IQR 7–18 months; Fig 2D). The interval between arrival in Germany and performance of medical screening (information available from 257 URMs, i.e., 28.9%) was short (S1 Fig)—a median of 3 days (range 0–120 days; IQR 2–4 days).

### Health complaints

Based upon history taking and clinical examination, dental problems represented the most frequent complaints and were present in 65.8% of all URMs. The single most frequent ID in the cohort was scabies (14.2% of URMs). Mental health issues (e.g., sleep problems, mood instability, signs of anxiety, suicidal tendencies, or substance use) were noted in 24.6% of URMs (Fig 2E).

## HBV and HIV infections

Study participants originating from countries with a high prevalence of chronic HBV infections (cutoff defined as ≥8%) were screened for hepatitis B antigen (HBsAg) [5,22]. Among the 776/890 participants tested, a total of 60 individuals (7.7% of all patients tested; Table 1) were identified with active HBV infection. Because most URMs were in our study area for a limited time period, we were only able to offer complete HBV diagnostics (including viral load) in 24/60 individuals (40.0%). In accordance with the European Association for the Study of the Liver (EASL) classification 2017 [22], the majority of tested URMs with active HBV infection were negative for hepatitis B e-antigen (HBeAg) ("inactive carrier"; i.e., 75.0%) and had low viral load in the blood ($<10^4$ IU/ml). Of note, the range of viremia differed substantially, ranging from <10 IU/ml to $1.2 \times 10^9$ IU/ml. In 5/26 of HBsAg-positive patients with known viral load, a viral load of $>10^6$ IU/ml was detected. Serum levels of aminotransferases did not correlate with viral load and often were normal in infected individuals (S2 Fig). Mode of transmission was unknown in all 60 patients. Most of the HBsAg-positive patients (15/21, 71.4%) had positive hepatitis A virus (HAV) serology indicative of previous HAV infection. No cases of combined HBV and HCV coinfection were identified. We did not initiate antiviral treatments in any of the patients with HBV infection, because the chronicity of infection could not be proven during the initial visit. Instead, the treatment mainstay was patient education regarding avoidance of hepatotoxic substances and sexual transmissibility of the disease. We also scheduled follow-up visits at our pediatric ID department and/or other appropriately specialized centers every 6 months.

HIV infection was detected in 3/760 (0.4%) of URMs, all originating from sub-Saharan Africa. Of note, in all three patients, HIV infection was detected through screening and not by means of clinically indicated testing. In these three patients with HIV, antiretroviral treatment was initiated after extensive patient counseling about the disease.

## Parasitic diseases

With a prevalence of 14.2% (*n* = 126) in the study cohort, scabies was the most frequent parasitic disease (Fig 2E), as well as the most frequent ID in the cohort overall.

**Table 1. Screening for hepatitis B (*n* = 776), and HIV (*n* = 760).**

| Characteristic | Number of patients | Percent of tested patients |
|---|---|---|
| **HBV status** (known for 776/890; 87.2%) | | |
| HBsAg positivity | 60 | 7.7% |
| Unknown mode of transmission | 60 | 100.0% |
| Phase of HBV infection [22] (known for 24/60 patients) | | |
| HBeAg-positive HBV infection ("immune tolerant") | 5 | 20.8% |
| HBeAg-positive HBV infection ("immune active") | 1 | 4.2% |
| HBeAg-negative HBV infection ("inactive carrier") | 18 | 75.0% |
| **HIV status** (known for 760/890 patients) | | |
| Anti-HIV IgG positivity | 3 | 0.4% |

All URMs coming from high-prevalence countries of origin (HBsAg prevalence ≥8%) were screened for HBsAg. The positive participants were then tested for HBeAg. Serological screening for HIV was conducted in all URMs originating from countries with HIV prevalence >1%. HBeAg, hepatitis B e-antigen; HBsAg, hepatitis B antigen; HBV, hepatitis B virus; HIV, human immunodeficiency virus; IgG, immunoglobulin G; URM, unaccompanied refugee minor.

**Table 2. Screening of parasite infection.**

| Characteristic | Number of patients | Percent of tested patients |
|---|---|---|
| **Eosinophil counts** (known for 871/890 patients; 97.9%) | | |
| normal (<500/µl) | 707 | 81.2% |
| 500–1,000/µl | 110 | 12.6% |
| >1,000/µl | 54 | 6.2% |
| **Stool parasites** (n = 119 samples) | | |
| None | 100 | 84.0% |
| Intestinal schistosomiasis | 8 | 6.7% |
| Giardiasis | 6 | 5.0% |
| Hook worm | 2 | 1.7% |
| *Taenia saginata* | 1 | 0.8% |
| *Strongyloides stercoralis* | 1 | 0.8% |
| Amoebiasis | 1 | 0.8% |
| **Urinary parasites** (n = 101 samples) | | |
| Urinary schistosomiasis | 6 | 5.9% |

Of 164 unaccompanied refugee minors with detected eosinophilia (>500/µl), stool tests were performed in 67.7% of cases. One to three stool samples per participant were tested. In the majority of cases (93.7%), two or three samples could be acquired. Besides eosinophilia, stool tests were performed when relevant gastrointestinal symptoms were present (n = 7). Coinfection of two stool parasites was rare (n = 1). In case of eosinophilia and negative stool parasitology (n = 94) and/or in patients with detected or reported hematuria (n = 4), microscopy of urine was performed.

Screening for stool parasites was performed in symptomatic patients (n = 7; 0.8%) and in patients with blood eosinophilia (n = 164; 18.8%). However, in a significant proportion of individuals with eosinophilia (32.3%, 53/164), stool samples could not be tested. In most cases, this was because the URMs became relocated to another part of Germany. Overall, the diagnostic yield of stool examinations was low. Only 19 pathogens could be detected in a total of 119 stool samples (16.0% positive tests), with intestinal schistosomiasis being the most frequent pathogen (n = 8). Six patients with urogenital schistosomiasis were identified via 4-hour midday urine microscopy. This was performed in response to the presence of microhematuria, for cases in which patients had a history of macrohematuria or dysuria without bacterial urinary tract infection, and in patients with blood eosinophilia along with negative stool investigation (Table 2).

As shown in Table 2, a significant number of URMs with eosinophilia tested negative for parasites (63/78, 80.8%), even when three stool samples were analyzed. Comparing eosinophil counts from patients with proven parasitic disease with those from patients with three negative stool samples and absence of hematuria revealed that eosinophilia was not specific for the presence of intestinal parasitic disease (p = 0.83, S3 Fig). This was not improved by raising the threshold to 1,000 eosinophils/µl, because only 7/19 (36.8%) patients with proven intestinal parasitosis had eosinophil counts of >1,000/µl.

## Tuberculosis

Of the 890 patients eligible for screening, the majority (n = 751/874; 85.9%) were screened radiologically, whereas IGRA screening was conducted in 76 (8.7%, Table 3). IGRA screening was performed in patients under 15 years of age (n = 43), as well as in a select number of patients (n = 33) over 15 years old, (e.g., in case of pregnancy or when X-ray diagnostics

**Table 3. TB screening.**

| Characteristic | *n* of patients | Percent of tested patients |
|---|---|---|
| **Screening modality (*n* = 874/890 screened patients; 98.2%)** | | |
| Chest X-ray only | 751 | 85.9% |
| IGRA only | 76 | 8.7% |
| Chest X-ray + IGRA[1] | 47 | 5.4% |
| **Abnormal TB screening[2]** | 75/874 | 8.6% |
| Abnormal chest X-ray | 34/798 | 4.3% |
| Positive IGRA | 58/123 | 47.2% |
| **Final TB workup** | | |
| Negative screening | 799/874 | 91.4% |
| Lost to follow-up | 11/874 | 1.3% |
| Latent TB infection | 38/123 | 30.9% |
| Pulmonary TB | 15/874 | 1.7% |

URMs originating from countries with high prevalence of TB (TB prevalence ≥20/100,000) were screened for TB. URMs <15 years of age were screened by IGRA, and URMs ≥15 years of age were screened by chest X-ray. In case of positivity, a second diagnostic mean (IGRA/chest X-ray) was added, and further diagnostics were performed. When active TB was suspected clinically[1], both tests were performed in parallel.[2] Abnormal TB screening was defined as IGRA positivity, X-ray abnormalities, or both.

Abbreviations: IGRA, interferon gamma release assay; TB, tuberculosis; URM, unaccompanied refugee minor

already had been performed in another European transit country). In 47 patients, both X-ray diagnostics and IGRA were initially performed because of clinical or radiological suspicion of active TB.

The rate of latent TB infection (LTBI) patients among the URMs screened by IGRA was high (38/123, 30.9%). Overall, we diagnosed 15 patients, all originating from sub-Saharan Africa, with pulmonary TB (1.7% of all screened URMs). Acid-fast bacilli only could be detected microscopically in the sputum of 3/15 TB patients (20.0%). Classical clinical symptoms (fever, cough >2 weeks, and weight loss) were neither sensitive nor specific for the identification of patients with active TB (S2 Table). Although more detailed analyses were hampered by the relatively small cohort of patients with pulmonary TB (*n* = 15) or LTBI (*n* = 38), we found that a positive history of cough for 2 weeks or longer was more common among TB patients than among URMs who had had a negative TB screening (26.4% versus 5.5%). However, coughing was also commonly reported among patients with a final diagnosis of LTBI (18.4%; S2 Table). The ratio of URMs who missed TB screening was low (*n* = 16; 1.8%), with half of them stating that they already had been screened with a chest X-ray elsewhere in Europe prior to coming to Germany. All patients with pulmonary TB were started on standard treatment regimens (isoniazid/rifampicin/pyrazinamide ± ethambutol) after cultural isolation of mycobacteria by sputum analysis, bronchoscopy, or endobronchial ultrasound-guided transbronchial needle aspiration (EBUS-TBNA) whenever possible. For LTBI patients, we recommended chemoprophylaxis (isoniazid ± rifampicin) in adolescents younger than 17 years or in older patients if treatment adherence (and safety regarding liver toxicity) was warranted.

## Discussion

Here, we present the results of a cross-sectional study on ID screening in a cohort of 890 URMs in Germany during 2016–2017. Scabies (14%) and chronic hepatitis B (8% of URMs from high-prevalence countries) were the most frequently identified IDs, whereas active TB

(1.7%) and HIV (0.4%) were diagnosed in a small minority of refugees. Importantly, the vast majority of IDs (apart from scabies) were not recognized clinically, either because the URMs were asymptomatic or else because the symptoms displayed were nonspecific. This underscores the need for symptom-independent screening approaches.

## Sociodemographics

We found that a majority of the URMs originated from Africa and entered Europe via the central Mediterranean route. The sociodemographic composition of our cohort differs from other studies [1,14]. In a retrospective study, Pohl and colleagues reported on 93 young patients (median age 5.7 years, originating mainly from Eritrea, Syria, and Afghanistan) who were hospitalized in a tertiary medical center in Switzerland [1]. In a study from southeast Germany, URMs described in an outpatient care setting were older (median age 16 years) and originated mainly from Somalia, Eritrea, and Afghanistan [14]. Our finding that approximately one-third of URMs underwent a journey lasting over 1 year underscores the potential for enormous physical and psychological impacts.

## HBV and HIV

The overall prevalence of HBsAg positivity within our cohort (7.7%) was high in comparison to other studies [10,12,13]. Because only URMs originating from high-prevalence countries (8%) were selected for screening [5, 19], a selection bias must be taken into account. This approach differs significantly from the American, Canadian, and Australian guidelines on ID screening in new immigrants, as these countries additionally recommend performing HBsAg screening in individuals arriving from countries with intermediate prevalence ($\geq$2%) [15,16] or else in all URMs [17]. In our cohort, the high number of HBV-infected URMs justified continuous screening. The high viral load present in some patients suggested an urgent need for patient counseling, especially with regard to sexual practices, as well as for close follow-up to confirm diagnosis and to avoid losing track of the patient. Vaccination of nonimmune people working and living together with URMs should be provided to prevent transmission. The best possible prevention strategy would be rigorous implementation of national vaccination programs in the URMs' countries of origin, as this is likely to demonstrably lower HBV prevalence [23]. The low number of URMs infected with HIV (0.4%) is in line with previous studies [14,24]. Despite this low HIV prevalence, HIV screening still should be offered to all refugee minors in order to minimize transmission risks [19]. Internationally, approaches to HIV screening of URMs vary. Some authorities recommend HIV screening for all refugee minors [16], whereas others screen only unaccompanied adolescents [17] or adolescents originating from countries with an HIV prevalence of $\geq$1% [15]. Because the number of URMs tested for HCV was low in our cohort, our data interpretation here was limited.

## Parasitic diseases

We found scabies to be the most prevalent ID (14.2%) in our cohort. However, because diagnosis was based upon clinical evaluation and additional diagnostic tests were not performed, disease prevalence may have been overestimated.

Overall, it was only in a low proportion of patients with eosinophilia that parasitic infection was able to be detected by stool and urine analyses (21/119, 21.0%). Although we cannot definitively rule out parasitic infections in participants with negative stool and urine diagnostics, our data support the possibility that eosinophilia testing may not be suitable for the purpose of identifying patients who require workup for parasitic disease [25–27]. However, because routine serological testing for schistosomiasis and strongyloidiasis was lacking, this may have led

to an underestimation of the helminth load and, therefore, may have skewed our data interpretation. Moreover, we may have missed some patients with asymptomatic malaria, although *Malaria tropica* could be ruled out due to the travel time after leaving endemic regions.

Our screening approach may explain this low diagnostic yield because only symptomatic individuals and those with eosinophilia >500/μl received additional testing (stool and urine examination), and no serologies for schistosomiasis or strongyloidiasis were performed. Some non-European authorities have suggested presumptive treatment of parasitic infection and a focus on specific serologies [15–17,28].

## Tuberculosis

The overall prevalence of active TB (1.7%) in our cohort was low. This was well in line with findings from other screening studies [29–31]. Most URMs with active TB originated from the Horn of Africa, and all were from sub-Saharan African countries. Most URMs with TB were asymptomatic. As other studies also have shown [31], clinical data relating to weight loss, fever, and chronic cough did not help identify those with active TB disease versus LTBI or TB-naive patients (S2 Table). This finding confirms the need for active screening in order to achieve timely TB diagnoses [32].

In our cohort, there were only three TB sputum–positive patients (0.3%)—a finding that reconfirms the minimal risk URMs represent for the domestic population with regard to onward TB transmission. As compared with other published data [32], the proportion of URMs lost to follow-up during the workup of pathological TB screening was low (1.3%).

The currently recommended screening approach [19] focuses on the search for active TB and enforces screening for LTBI only in those <15 years of age. This recommendation is based upon data showing that young children with LTBI are at higher risk for developing active TB disease [33]. This elevated TB risk is less likely in adolescents and adults. In Europe, compliance and chemoprophylaxis completion rates among migrants with LTBI identified via screening have been reported to be low [34]. When treatment adherence is low, necessary follow-up visits get missed, regular blood tests are not regularly performed, and treatment safety is reduced. In such circumstances, LTBI screening may seem unjustified. However, in the subgroup of URMs <15 years old, the LTBI rate reached 23.3% in our study. For this age group, LTBI screening appears to be medically justified.

Our study contains certain limitations: For parasite screening, no serology-based testing was performed; thus, some infections may have been overlooked. Moreover, the TB screening approach was a mixture of screening for active TB in older URMs as well as LTBI in younger URMs. Therefore, true LTBI prevalence remains unknown for our study cohort. With the exception of HBV and HIV, screening for sexually transmitted diseases was not included in our study. Because of the small number of female URMs, no conclusions regarding health state differences between the sexes can be stated.

Most importantly, our results cannot be generalized to the overall child and adolescent refugee population and/or to other areas of Europe, because the characteristics of accompanied refugee minors are likely to differ given the diverse countries of origin and ethnicity of refugees across Europe [8].

As we have shown, our screening approach was practicable and had relatively low drop-out rates. However, some changes should be considered, especially in screening for parasitic diseases. Blood eosinophilia detection, together with stool and urine investigation, pose logistical challenges. They are also expensive and provide relatively unreliable results [35]. Our data indicate that a general screening for blood eosinophilia as a test for parasitic diseases is not immediately helpful. For this reason, studies are needed that address presumptive parasitic

treatment or a combination of serology for schistosomiasis and strongyloidiasis [27,28,35,36] with symptom-based treatment in pediatric refugees. Given the unreliable data on disease prevalence in conflict zones, screening of all children and adolescents for active TB, hepatitis B, and HIV should be considered, especially if the numbers of migrants arriving in Europe continue to go down. Of note, the evaluated screening approach [19] was intended for a situation in which an unusually high number of children were migrating to Germany, as was the case during 2015 and 2016. Hence, our screening approach was restricted as compared with other established screening procedures [15,16,17]. Screening modalities for refugees and migrants inside the European Union need to become more clearly defined. For this, European evidence-based guidelines are urgently needed [15,18]. We plan to use the data from this study in order to revise screening recommendations—especially with respect to screening for parasitic infections.

In addition, in most cases, we had little (if any) reliable information from our patients regarding previous screening investigations before they arrived in Germany. Many of our URMs had healthcare contacts at their first ports of entry into Europe. This suggests that at least some of the investigations are likely to have been repeated and therefore could have been avoided. Some patients with active TB reported probable tuberculostatic treatment in other European countries before arriving in Germany. Unfortunately, however, their treatment was discontinued before completion, usually because the patient migrated or became transferred to another country. This suggests that the need for inter-European healthcare communication regarding these patients is urgent. Electronic health records would be ideal. In order to improve upon the follow-up of refugees with chronic infections, as well as to reduce frequency of repeat tests and thus rationalize screening and healthcare procedures, a more efficient transfer of healthcare records and information is needed.

## Supporting information

**S1 STROBE Checklist. Checklist of items that should be included in reports of cross-sectional studies.** STROBE, Strengthening the Reporting of Observational Studies in Epidemiology.
(DOC)

**S1 Text. Electronic questionnaire for data acquisition (translated English version).** The questionnaire was completed by the physician performing the screening visit.
(PDF)

**S1 Fig. Delay of screening after arrival in Germany in unaccompanied refugee minors within the cohort; indicated is the screening time point in days after arrival to Germany.**
(EPS)

**S2 Fig. Distribution of aminotransferase levels in HBsAg-positive patients.** Normal values are depicted with dotted lines. HBsAg, hepatitis B antigen.
(EPS)

**S3 Fig. Comparison of eosinophil distributions in patients with or without detected stool parasites.** Only patients with >500 eosinophils/μl and three analyzed stool samples were included. Analyzed by unpaired $t$ test (Welch correction).
(EPS)

**S1 Table. Countries of origin of unaccompanied refugee minors within the cohort ($n$ = 890).**
(DOCX)

**S2 Table. Clinical warning signs for active tuberculosis infection in screened unaccompanied refugee minors (*n* = 874).**
(DOCX)

## Acknowledgments

We would like to thank the pediatric nurse Christina Kronthaler and study nurse Bianca Rippberger for their valuable support. The article processing charge was funded by the University of Freiburg in the funding program Open Access Publishing.

## Author Contributions

**Conceptualization:** Markus Hufnagel, Roland Elling.

**Data curation:** Kristin Eder, Nadine Binder, Jan Rohr, Roland Elling.

**Formal analysis:** Ales Janda, Kristin Eder, Roland Fressle, Anne Geweniger, Natalie Diffloth, Nadine Binder, Jan Rohr, Markus Hufnagel, Roland Elling.

**Investigation:** Ales Janda, Kristin Eder, Roland Fressle, Anne Geweniger, Ana-Gabriela Sitaru, Philipp Henneke, Markus Hufnagel, Roland Elling.

**Methodology:** Anne Geweniger, Natalie Diffloth, Maximilian Heeg, Ana-Gabriela Sitaru, Jan Rohr, Philipp Henneke, Markus Hufnagel, Roland Elling.

**Project administration:** Natalie Diffloth.

**Software:** Natalie Diffloth.

**Supervision:** Philipp Henneke, Markus Hufnagel.

**Validation:** Maximilian Heeg.

**Visualization:** Maximilian Heeg, Roland Elling.

**Writing – original draft:** Ales Janda, Philipp Henneke, Markus Hufnagel, Roland Elling.

**Writing – review & editing:** Ales Janda, Kristin Eder, Roland Fressle, Anne Geweniger, Maximilian Heeg, Nadine Binder, Ana-Gabriela Sitaru, Jan Rohr, Philipp Henneke, Markus Hufnagel, Roland Elling.

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
