## [Decision Letter · Decision Letter 0]

19 Nov 2019

Dear Dr. Elling,

Thank you very much for submitting your manuscript "Comprehensive infectious disease screening among a cohort of 890 unaccompanied refugee minors in Germany from 2016 to 2017" (PMEDICINE-D-19-03543) for consideration at PLOS Medicine for our upcoming special issue on refugee and migrant health. 

Your paper was discussed among the editorial team and sent to independent reviewers, including a statistical reviewer. The reviews are appended at the bottom of this email and any accompanying reviewer attachments can be seen via the link below:

[LINK]

In light of these reviews, we will not be able to accept the manuscript for publication in the journal in its current form, but we would like to invite you to submit a revised version that fully addresses the reviewers' and editors' comments. You will appreciate that we cannot make a decision about publication until we have seen the revised manuscript and your response, and we expect to seek re-review by one or more of the reviewers. 

We hope to receive your revised manuscript within two weeks. Please email us (plosmedicine@plos.org) if you have any questions or concerns.

Please let me know if you have any questions. Otherwise, we look forward to receiving your revised manuscript soon. 

Sincerely,

Richard Turner, PhD

rturner@plos.org

Please provide a fuller account of the ethical situation, in which you discuss the responsible party for unaccompanied minors; the issue of informed consent not having been obtained; and the accountability for providing appropriate treatment for the diseases identified. 

Please remove "890" from your title, and add a study descriptor to the title following a colon, e.g. "... : a cross-sectional study". 

We ask you to combine the "aims" subsection of your abstract with the preceding "background" subsection. 

Please combine the "methods" and "findings" subsections of your abstract, and add a new final sentence to the new combined subsection summarizing the study's main limitations. 

Please make that "sociodemographic" at line 35, and "January" at line 38. 

At line 48, please start the sentence with "In this study, we found that ..." or similar. 

After your abstract, we will need to ask you to add a new and accessible "author summary" section in non-identical prose. You may fine it helpful to consult one or two recently published research papers in PLOS medicine to get a sense of the preferred style. 

Please remove the text at line 104-107, or move it to the discussion section. 

Early in the methods section of your main text, please state whether or not the study had a protocol or prespecified analysis plan, and if so attach the relevant document(s) as a supplementary file. Please highlight analyses that were not prespecified. 

The first paragraph of your discussion section should summarize the study's findings, and we ask you to expand the text at lines 293-297 to achieve this. 

Please remove trade marks throughout your ms. 

Please add a completed checklist for the most appropriate reporting guideline, e.g., STROBE, as a supplementary document, and refer to this early in your methods section. In the checklist, individual items should be referred to by section (e.g., "Methods") and paragraph number rather than by line or page numbers, as the latter generally change in the event of publication. 

Comments from the reviewers:

*** Reviewer #1: 

I confine my remarks to statistical aspects of this paper. These were quite simple, but appropriately so. However, they were not adequately described in the methods section. The "statistical analysis" section is now 3 lines long. It needs to say what was done and how it was done, even if it's simple. It also needs to describe how variables were operationalized. E.g. what were the "3 routes" to Europe? How were people assigned to one of them? What is "nutritional status" (from the results, it seems like it is BMI ... which isn't really a measure of nutrition, it's a rather poor measure of obesity but a person can be obese and malnourished)/ This sort of thing should be done for all the variables.

More specific comments:

Lines 92-93 These adjectives "limited number" and "select population" apply to all studies including this one. Please give details and say why your paper is better.

Lines 163-164 I'm a little confused here. What is "documented date of birth"? How was it determined? What about undocumented DOB - just ask the person their birthday?

Line 166-167 Certainly 6.0% girls seems low, but how do you know it is "underrepresented"? It's at least theoretically possible that nearly all URMs are boys. Do you have evidence one way oir the other?

Line 175 and 178 Please give interquartile range, as well

Figure 2 D - This isn't really a good graph. Why was this unusual format chosen? Maybe just text about this (it's only 3 pieces of information).

Peter Flom

*** Reviewer #2: 

Overall this is an important descriptive paper for a large population of URM who live in Germany and throughout the EU.

Introduction:

- 2nd & 3rd lines of 1st paragraphs would be strengthened by: removing the 2nd line or adding what kinds of challenges: economic, educational, health etc. For the 3rd line, I would recommend reframing to the fact that infectious diseases have varying endemicities globally rather than framing it as it is reasonable to assume that the endemicity is higher in place where refugees are migrating from.

-Please more explicitly state the research question/objective

Methods:

-Please describe the types of patients cared for at a private practice in southwest Germany. 

-Figure E. Please revise "mental problems" to say "mental health concerns"

Results:

Very well described, thorough discussion. Well used tables and figures.

Discussion:

It would be helpful to further describe a brief line or two on how the study is similar or different to Switzerland and SE Germany as described, in demographics, screening or ID prevalence?

Incorporation of the findings of this paper on Hepatitis B by Yun et al includes a large cohort of youth from E. Africa and would be a fitting for comparison, in addition to their discussion on immunization.

Increasing Hepatitis B Vaccine Prevalence Among Refugee Children Arriving in the United States, 2006-2012.

Yun K, Urban K, Mamo B, Matheson J, Payton C, Scott KC, Song L, Stauffer WM, Stone BL, Young J, Lin H.

Am J Public Health. 2016 Aug;106(8):1460-2. doi: 10.2105/AJPH.2016.303203. Epub 2016 Jun 16.

The findings of eosinophilia not being predictive of parasitic disease which supports the rationale has been described in a few studies in North America that would further strengthen the discussion on this topic, and may suggest the need for an alternative approach to parasitic disease screening vs. presumptive treatment among URM in Germany. Potential Papers to consider:

 Eosinophilia: A poor predictor of Strongyloides infection in refugees.

Naidu P, Yanow SK, Kowalewska-Grochowska KT.

Can J Infect Dis Med Microbiol. 2013 Summer;24(2):93-6.

Eosinophilia and the seroprevalence of schistosomiasis and strongyloidiasis in newly arrived pediatric refugees: an examination of Centers for Disease Control and Prevention screening guidelines.

Dawson-Hahn EE, Greenberg SLM, Domachowske JB, Olson BG.

J Pediatr. 2010 Jun;156(6):1016-1018.e1. doi: 10.1016/j.jpeds.2010.02.043. Epub 2010 Apr 18.

High prevalence and presumptive treatment of schistosomiasis and strongyloidiasis among African refugees.

Posey DL, Blackburn BG, Weinberg M, Flagg EW, Ortega L, Wilson M, Secor WE, Sanders-Lewis K, Won K, Maguire JH.

Clin Infect Dis. 2007 Nov 15;45(10):1310-5. Epub 2007 Oct 11.

The discussion of the strong need to weave together surveillance systems and results was well described in a recent paper about migrants in the UE by Kunst et al and team and would further strengthen the basis for your argument in this area:

Int J Tuberc Lung Dis. 2017 Aug 1;21(8):840-851. doi: 10.5588/ijtld.17.0036.

Tuberculosis and latent tuberculous infection screening of migrants in Europe: comparative analysis of policies, surveillance systems and results.

Kunst H1, Burman M1, Arnesen TM2, Fiebig L3, Hergens MP4, Kalkouni O5, Klinkenberg E6, Orcau À7, Soini H8, Sotgiu G9, Zenner D10, de Vries G11.

*** Reviewer #3: 

PLOS Medicine Review PMEDICINE-D-19-03543

Comprehensive infectious disease screening among a cohort of 890 unaccompanied refugee minors in Germany from 2016-2017

Thank you for the opportunity to review the above manuscript. The authors outline infectious diseases screening on a large cohort of predominantly male adolescent unaccompanied refugee minors (URM) presenting in transit to a paediatric practice in South West Germany. Given the lack of symptoms in many URMs, their finding appear to support universal infection screening regardless of clinical symptoms. It should be noted however that the clinicians did not complete full screening when compared to other published international refugee health management guidelines but an abridged version (local German guidelines). 

The comments below are intended to improve the clarity of the manuscript and thus the message for the wider readership.

MAJOR COMMENTS

(1) It would be useful to comment upon the structure of the paediatric practice (which undertook screening) and how the infectious diseases screening of the URM was incorporated into the holistic health assessment of the cohort. Specifically, how were the patients managed with respect to treatment of identified diseases/infections and subsequently followed up?

(2) The authors state that the German recommendations were not based on local data; the Australian (2009 and revised 2016; Chaves et al) and Canadian (2011 Pottie et al) guidelines are evidence based guidelines with specific details relating to epidemiology of similar cohorts screened (e.g Mutch et al 2012 Journal of Paediatrics and Child Health, Paxton et al 2019 JPCH). Given this fact, I am querying why malaria was not included in the screening protocols, particularly as many URMs appeared to originate from (or transit through) malaria endemic regions. This would also be important to consider in your findings of peripheral eosinophilia and is a limitation of this paper. 

(3) Was broader STI screening considered in this cohort (e.g. HCV, syphilis, chlamydia, gonorrhoea)? This is a high risk/vulnerable population (especially for the small percentage of URM females): other intentional screening recommendations include routine testing of this cohort due to heightened risks related to gender based violence, which are often unreported. 

(4) The authors stated that 'no antiviral treatment was instituted" (Results section HBV and HIV infection line 210) - does this specifically apply only to the cohort that were HBV infected or inclusive of the HIV positive patients? If no treatment was commenced, how was this clinically justified for the HIV cohort? What follow-up, counselling and referrals were put in place?

(5) Did the practice undertake review of immunisation status (e.g. rubella, varicella, HBV, HPV)? What catch up immunisation processes were put in place for this cohort? Specifically for the HBV non-immune cohort, how did you commence catch up regimens (in the context of high prevalence of HBV infected URMs in this cohort)?

(6) It is unclear how nutritional status was ascertained. Was this based on normal haemoglobin and MCV levels on the FBP and normal BMI status? Were iron and vitamin D levels assessed in this cohort? There are data pertaining to children with LTBI who have low vitamin D levels, having an increased risk of progression to active TB. Given that 30% of the cohort with an IGRA were found to have LTBI, this is an important consideration. Were (1) vitamin D levels in this cohort and (2) how did you institute prophylactic or active therapy in this cohort?

(7) I note that almost a quarter of the cohort have "mental health" symptoms at first assessment. How were the wider components of refugee health assessment (e.g. education, dental health, immunisation, trauma, malnutrition, haemoglobinopathy screening, non-communicable diseases etc) dealt with in this practice or were they all referred on to other providers? 

(8) How do the German guidelines align with the international guidelines (Aust/Canada) which recommend routine screening for all refugee children and adolescents across all of the above domains (not just infectious diseases focused)?

(9) What changes to clinical practice guidelines will be (or have been implemented) as a result of these findings? This is an important aspect of this study to include in your discussion, including whether broader screening should be implemented e.g helminth serology, other clinical domains (in keeping with other international processes).

MINOR COMMENTS

(1) Limitations; the lack of malaria testing and serology for schistosomiasis and strongyloidiasis is am important consideration and likely underestimates the helminth load demonstrated in your results. Many refugees are asymptomatic with respect to infection; this would also impact on your assessment of eosinophilia (used as an adjunct to serology). 

(2) What proportion of the URMs had a protracted refugee transit (> 5 years)? Did this have any bearing on the prevalence of infections (e.g. scabies, malnutrition, dental caries, HBV)?

(3) Some typographical errors are noted in the paper; particularly in the abstract. It would also be better to use the term "practical" rather than "practicable" for ease of reading. 

(5) How did you address language and literacy barriers within the screening process? 

(6) In your introduction, the authors state that chronic parasitic infections may lead to problems such as anaemia, nutrient deficiency or stunting. What was the correlation in this cohort?

(7) Where there any differences in the infection profiles between female and male URMs?

(8) Data analyses (plural) is the correct term; use of analysis incorrectly through the manuscript is noted.

(9) Age distribution is skewed (ie non-normally distributed). Median and interquartile range would be more appropriate to use rather than mean age. 

(10) The authors are incorrect in stating that no patients were connected with HCV as it was only tested for in a very small subset (21 patients). This should be corrected and noted in limitations. 

(11) Looking at BMI status of patients with respect to TB status, it appears non-significant; can the authors confirm this interpretation?

***

[LINK]

---

## [Decision Letter · Decision Letter 1]

3 Feb 2020

Dear Dr. Elling,

Thank you very much for re-submitting your manuscript "Comprehensive infectious disease screening among a large cohort of unaccompanied refugee minors in Germany: a cross-sectional study" (PMEDICINE-D-19-03543R1) for consideration at PLOS Medicine.

I have discussed the paper with editorial colleagues and the guest editors for the special issue, and it was also seen again by two reviewers. I am pleased to tell you that, once the remaining editorial and production issues are dealt with, we hope to be able to accept the paper for publication in the journal.

[LINK]

Please let me know if you have any questions in the meantime. Otherwise, we look forward to receiving your revised manuscript shortly. 

Kind regards,

Richard Turner, PhD

rturner@plos.org

Requests from Editors:

Please remove the word "large" (cohort) from the title; and add a date range for when the study was done. 

Please also remove "large" from line 35.

At line 47, please make that "criterion". 

At line 56, please make that "were detected clinically" (assuming this is correct). 

We suggest noting the prevalence of mental health disorders in your abstract. 

At line 64, please substitute the word "fleeing" with "migrating" or similar. 

At lines 79 and 388, would adapting the wording to "... were not detected clinically." be appropriate?

At line 83, please remove the word "completely". 

Please adapt the sentence at line 115 to make it clear that the prevalence is lower in Germany (e.g., "In Germany ... a twenty-fold lower prevalence."). 

At line 162, please make that "non-prespecified". 

At line 169, please substitute "transit" (or similar) for "flight". 

At lines 170 and 173, please substitute "drug [substance] use" (or similar) for "drug [substance] abuse". 

Please reword the text at line 209 to make it clear whether the legal guardian or municipality provided consent. Please make it clear whether consent was informed, and how it was established (written/verbal). The wording in the text seems to contradict the note in your response to editorial points that it was "not possible to prospectively obtain informed consent" and we ask you to make the actual situation clear in the revised ms. 

At line 213, please make that "Data analysis was performed in an anonymized fashion." (assuming this is correct). 

At line 234, we suggest making the information less specific, e.g. "... the youngest ... was 9 years of age".

At line 268, can you be more specific about "mental health issues", i.e., were disorders noted?

At line 282, should that be "anti-HBV immunity"?

At line 387, please amend the text to "... were diagnosed in a small minority of refugees" or similar. 

At line 388, please make that "were not recognized clinically". 

At line 389, please adapt the text to "because they showed" or similar. 

At line 399, please substitute "endure an escape" with "underwent a journey" or similar, and remove the word "flight". 

At line 413, please avoid the word "subjects" (in favour of "people" or similar). Would "provided" be a good substitute for "vigorously controlled"?

At line 420, please substitute "countries" or similar for COI. 

At line 435, please expand on "M.tropica", as this does not appear to occur earlier in your ms. 

At line 452, we suggest revisiting the text "reconfirms the minimal risk". It may be that noting the small proportion of sputum positive patients could suggest a low risk of onward transmission. You might wish to briefly mention local policies on BCG vaccination, which could be relevant to this point. 

At line 462, would that be "... reached 23.3% in this study"?

At line 483, please substitute "migrating" for "immigrating". 

At line 485, should reference citations be substituted for "(Pottie; CDC; Chaves)"?

In reference 20, please abbreviate "World Health Organization" as "WHO", or not at all. 

Please revisit reference 31, as the title appears to be partially duplicated. 

In figure 2, please remove the word "escape" (route to Europe). 

Comments from Reviewers:

*** Reviewer #1: 

The authors have addressed my concerns. However they wrote

<<<

We agree with reviewer 2 that the representation is unconventional and changed Figure 2D to a more classical graph (pie chart).

>>>

I'm not a fan of pie charts. See my blog post https://statisticalanalysisconsulting.com/graphics-for-univariate-data-pie-is-delicious-but-not-nutritious/

If the editors want to publish as is, that's OK, but I'd rather delete the pie chart and just describe the data (it's only 3 points of information)

Peter Flom

*** Reviewer #2: 

appreciate the authors additional consideration of the editors and 3 reviewers comments. I think altogether they have strengthened this paper. This paper is important as very little is understood about URM populations globally, however, I think the paper still requires some additional revisions for clarity and placing the papers in the context of the existing literature. Some additional suggestions follow:

-Reviewer 1 asked a question about the documented date of birth, and it was answered in a way as to clarify this question. The authors response should be added to the paper for clarity for the reader.

-Reviewer 2 recommended adding additional citations to the text. It's noted that these citations were added, however, additional sentences were not added placing this new study in the context of these previous studies. That component is very important. Additional comparison and potential emphasis or disagreement with these prior papers is needed to place this new study in context. 

-Reviewer 3 queried about STI testing. Given this study is focusing on ID it is important that this discussion and additional commentary from the authors is included in the discussion of this topic. Many unaccompanied refugee minors have experienced trafficking and are at high risk for it. Like many adolescents, they may not reveal their sexual history particularly for non-consensual sex, or that differs from their definition of sex. Arguably they are at high risk and more discussion of STI testing should be included in the discussion as well as acknowledged in limitations.

-Reviewer 3 raised a very important concern about language and literacy barriers. In the body of the text the more detailed description of the interpretation should be added as well as the use of interpreter rather than translator.

-Reviewer 3 raises an excellent point about chronic parasitic infection contributing anemia, nutrient deficiency or stunting being included in the introduction this sentence would benefit from a citation. Further, the description provided by the authors should be included in the paper. Please change infest to infect, throughout this comment and the paper

-Reviewer 3 raises an important point about the data as it relates to the few female refugees in the study. Please add your response to the comment to the limitations section. 

Introduction:

-For line 99, "...it seems reasonable to screen refugees for ID that bay me more prevalent in this population." Please provide more context for this assertion. I imagine that the authors mean that ID is higher prevalence in low and middle income countries and more refugees originate from these countries, however, I think this additional context is needed.

-In the conclusion to paragraph 1, I think it is important to also include that limited/no access to care may also contribute to delays in diagnosis

-Paragraph 2: It would be helpful to provide some context as to why Europe and Africa as continents are being compared specifically. I imagine this is due to the high number of URM from Africa. Further, please add language to reflect that these difference in prevalence relate to being low and high resource places rather than something inherent to the continents.

Methods:

-It is not clear what language the survey was written or administered in and if it was interviewer/interpreter assisted in completing this. were the questions developed with language and culturally concordat team members? Please provide additional context here.

Results:

-Please remove line 234-235 about the youngest study team member as this may be identifying.

-Figure 2: There are errors in the key as I believe C and E are flipped. Additionally it may be fitting to call it systems of clinical signs and symptoms, or systems and diagnoses given it relates to body systems and then scabies

-For Line 276-277, the majority should be placed in the context of the fact that it was only patients among the 24 that had additional testing. This would not reflect the majority overall, and is therefore misleading.

-Table 1, please move the section related to the 24 of 60 patients to another table or to another column as it is important that the % are interpreted to be reflective of the 24 patients not the overall 776 that had testing for HBV

-Line 341 please remove the frame "surprisingly" high, since this is the results section it should not include such descriptive words and opinions of the authors. Similarly please remove interestingly.

-Line 343 Please remove the line that TB patients were not contagious, or elaborate on the context. Clinicians commonly say that anyone with active disease is considered contagious.

-Line 346-347: Please remove the idea that there was a tendency toward lower BMI in TB patient as p=0.6 is far from significant.

-Paragraph 375-381 may fit best in the methods section as it does not seem like it follows from the results as it is not about something that was screened for

Discussion:

-Line 399-400 please add the citation that the finding is in reference to.

-Line 403-404 please provide additional context as to why it si worth noting that the URM are predominantly of African origin. Is it similar or different from prior studies? 

-Line 410-413: Please revise these lines or remove them as it is inaccurate as currently framed. The viral load being high in 20% of patients is 5 patients given that there were only 24 patients who had their viral loads evaluated. This is 0.6% of the population of those tested for HBV. Additionally, it would strengthen the discussion to add that HBV may be acquired through vertical transmission, blood or sex. It would be helpful to provide some additional context around what other forms of prevention might be helpful to consider as compared to large scale prevention in the country of origin

-Line 455-456 please include a citation for the recommendation around LTBI vs. active TB disase screening

-Line 475-476 Please add context as to how these findings are similar or different than the studies cited

-I recommend removing Table 4, the sample size is <5 in the majority of the cells (making it difficult to interpret), and individuals with latent TB do not have symptoms suggesting that those that have cough, weight loss or fever have another illness.

***

[LINK]

---

## [Editor Report · Decision Letter 2]

25 Feb 2020

Dear Dr. Elling, 

On behalf of my colleagues and the academic editor, Dr. Paul Spiegel, I am delighted to inform you that your manuscript entitled "Comprehensive infectious disease screening in a cohort of unaccompanied refugee minors in Germany from 2016 to 2017: a cross-sectional study" (PMEDICINE-D-19-03543R2) has been accepted for publication in PLOS Medicine. 

PRODUCTION PROCESS

PRESS

PROFILE INFORMATION

Thank you again for submitting the manuscript to PLOS Medicine. We look forward to publishing it. 

Best wishes, 

Richard Turner, PhD

Senior Editor 

PLOS Medicine

plosmedicine.org